

# RpNGS: an automated platform for pathogen identification and monitoring in clinical metagenomics data

Jing Zhou[1], Yao Tian[1], Min Yang[1], Ting Hao[2], Jun Ma[2,3,4] and Shengyu Wang[1]

[1] Department of Intensive Care Unit, The First Affiliated Hospital of Xi'an Medical University, Xi'an, Shaanxi, China
[2] Shaanxi Lifegen Co., Ltd, Xi'an, Shaanxi, China
[3] Guangdong Provincial Key Laboratory of Microbial Safety and Health, Institute of Microbiology, Guangzhou, China
[4] School of Food Science and Engineering, Shaanxi University of Science and Technology, Xi'an, Shaanxi, China

## ABSTRACT

**Background**. The capacity of metagenomic sequencing-based diagnostics to fully identify infections have made them useful instruments in clinical practice. We introduce an interactive platform that runs on a local server-class hardware resource and implements a number of open-source programs.

**Results**. RpNGS integrates an interactive tabular interface for the management of experimental processes, patient metadata, and automated sequencing analysis. This technology optimizes clinical reporting by autonomously generating standardized reports in Word format. We have utilized the platform on an artificial microbial community reference panel and several clinical metagenomics datasets from public databases to demonstrate the efficacy of this workflow.

**Conclusions**. RpNGS is an innovative, user-friendly standalone application designed to store laboratory data (including reagents, primers, contaminants and run configurations), manage clinical metadata, process FASTQ files and produce analytical and comparative reports (including Word documents) that can be readily reviewed and certified. Its interactive interface necessitates no programming expertise, rendering it an invaluable instrument for clinical metagenomic pathogen identification.

Corresponding authors
Jun Ma, nwu_majun@hotmail.com
Shengyu Wang,
wangshengyu@yeah.net

# BACKGROUND

The bacteria, fungi, viruses and parasites lead to virous infectious disease (*Ma, Yang & He, 2024*; *Kan et al., 2024*). In contrast to traditional molecular diagnostic tests like direct or multiplex PCR, clinical metagenomics utilizing next generation sequencing (mNGS) possesses the ability to identify all possible pathogens (*Zeng et al., 2022*). Currently, laboratory-developed mNGS is increasingly utilized in the diagnosis of infectious diseases (*Liu et al., 2021*; *Diao et al., 2023*; *Feng et al., 2024*). With the broad and successful application in clinical pathogen detection, serval bioinformatics software has been developed for mNGS data analysis. For instance, Kraken2 suite (*Lu et al.,*

*2022*), IDseq (*Kalantar et al., 2020*). However, bioinformatics pipeline is used for mNGS analysis, usually without user-friendly interface, included a number of different algorithms, developed for research purpose, focused on sequencing data analysis and constantly updated by software developers (*Chiu & Miller, 2019*; *Dhungel et al., 2021*). The purpose of this study was to provide a standardized workflow available to the public (https://github.com/mj200921059/RpNGS) that would make the clinical metagenomics application easier to execute, especially for non-expert users. The expert user can also change the summary information table of general tab interface and switch version or programs of the sequencing data analysis pipeline including classification algorithm and reference databases.

Using Shiny, various R libraries, and Conda environment setup with *Fastp, Bowtie2, Kraken2* and *Bracken*, we constructed RpNGS to generate consistent analysis tar reports to our partners. Unlike other apps, our workflow is a web application to gather the information among sample processing, nucleic acid extraction, library preparation, patient's biological data and mNGS data analysis, which can be readily launched by users with minimum expertise in shiny or R. Additionally, it features a set of strong and well-designed interactive plot and table for visualization and clinical reports preparation.

## MATERIALS & METHODS

### RpNGS software overview

RpNGS was written in R programming language (https://www.R-project.org) as a modular Shiny app (*Chang et al., 2024*), an R package for constructing interactive web applications. The web interface of this application was developed using *shiny*, *shinyFiles*, *shinycssloaders*, *shinydashboard*, *shinyWidgets*, and *shinyBS* libraries. This interface runs in a web browser and provides dynamic changes lab data and clinical metadata through interactive JavaScript tables powered by the DT package. In addition, RpNGS also delivers a set of sophisticated and well-designed interactive visualizations based on the plotly and leaflet package.

### Analysis workflow

A typical clinical metagenomic next-generation sequencing bioinformatics pipeline usually comprises quality control, host reads removal, taxonomic categorization and validation (*Miller et al., 2019*). The RpNGS analysis methodology automates metagenomic processing in a series of replicable stages by click "Confirm and Analyze" button after loading raw FASTQ files of one sequencing run into working directory and fill the related lab data.

Given these processes are conducted by R scripts that connect with external tools within a Conda environment (Fig. S1). The preparation of analysis workflow involves the Conda environment set-up and reference databases. We firstly constructed a *Fastp* conda environment and installed *Fastp*, *Bowtie2*, *Kraken2*, and *Bracken* programs before executing analysis pipeline (*Lu et al., 2017*; *Chen et al., 2018*; *Wood, Lu & Langmead, 2019*; *Bush et al., 2020*). The Bowtie2 index of each microbes was built by the bowtie2-build command with default parameters, while the reference database (PlusPFP.tar.gz) of kraken2 and bracken and human bowtie2 index were download from their websites

(https://benlangmead.github.io/aws-indexes/k2 and https://bowtie-bio.sourceforge.net/bowtie2/index.shtml, respectively).

The backend computational pipeline can be conducted within an R environment or delivered on a server for automated processing. The analytic pipeline utilizes *Fastp* for getting high-quality clean FASTQ file with read length >36 bp. Considering the genomic size disparity between humans and microorganisms, the majority of metagenomic read sets pertained to human DNA, even in samples with a similar number of cells. To better focus on the microbe's identification, the host reads should be removed before run classification analysis (*Bush et al., 2020*). Therefore, the cleaned FASTQ file are aligned to the hg37dec_v0.1 reference genome using *Bowtie2* with -U, –very-sensitive parameters to obtain the filtered data. Reads categorization and abundance estimate is independently created by *Kranken2uniq* with *–minimum-hit-groups 3* and *Bracken* with *-r 50 -l S* against a reference database. After that, we mapping taxonomy list to our pathogen list to acquire the potential pathogens.

In order to exclude the background influences such as microbial nucleic acid residing in reagents, sampling and lab settings, RpNGS construct a z-score for each species. The taxon with z-score >1 was classified as actually detected, which suggests the abundance of species in sample is higher than the controls. The z-score approach was first described in *Wilson et al. (2018)* and is applied in the IDseq and Pavian metagenomic platform (*Breitwieser & Salzberg, 2020*) to reflect the significance of relative abundance estimations in a sample as compared to the water controls. The z-score value of each taxon in one sample is calculated as follow:

$$z = \frac{x - \mu}{\sigma}$$

where $x$ stands for rpm of taxon in samples, $\mu$ is the average of rpm of taxon in control samples, and $\sigma$ means the of rpm of taxon in control samples.

Reads for candidate pathogen are independently extracted from the filtered FASTQ file of each sample. To validate the pathogen detection results, we run *Bowtie2* to map the pathogen specific reads to its matching reference genome for coverage detection, which illustrate by *Gviz*, *Rsamtools* and *GenomicAlignments* R packages in the third tab of RpNGS. The pathogen with greater mapped reads but less coverage will be removed from candidate pathogen list.

## Report workflow

R packages *flextable* and *officer* were used to generate a pathogen report each sample by exporting a reactive table from shiny into a pre-existing word template.

## External benchmarks-datasets and metrics

To evaluate the efficacy of in-house pathogen identification pipeline in RpNGS, we benchmarked it to previously published metagenomic pathogen identification results (*Diao et al., 2023*). A total of 23 respiratory pathogens at varied concentrations were individually mixed to create 14 microbial mixtures (samples S1–S13 and one negative control), which were then disseminated to 122 laboratories as part of a multicenter mNGS

quality assessment study (see Table S1). The raw FASTQ files from these samples were deposited in the Genome Warehouse of the National Genomics Data Center under project PRJCA015554. For benchmarking, we evaluated a total of 28 raw FASTQ files generated by lab005 using IDseq, with the host filter set to "human" and the background set to "none" (see Table S2). The IDseq was selected because it is well-known cloud-based, open-source bioinformatics platform developed for clinical metagenomic study. The sequence analysis procedure of IDseq platform involves three primary steps: host filtering and QC, assembly-based alignment, and taxonomy reporting and visualization. The computed metrics including the precision (P), the recall (R), and the F-score (F1) were calculated based on below formula: $P=TP/(TP+FP)$, $R=TP/(TP+FN)$, and $F1=2/((1/R) + (1/P))$, where TP are true positives, FP stands for false positives, FN represents false negatives.

### Application for pathogen identification

For the comparison with IDseq, we additionally downloaded real clinical metagenomics sequencing files of CHRF0000, CHRF0094 and CHRF0002 from the NCBI under BioProject PRJNA516582 and re-analysis on our in-house system. Samples CHRF0000, CHRF0094 and CHRF0002 in the original study, which contained 91 CSF samples and six water control (*Saha et al., 2019*), are a water control, *Streptococcus pneumoniae*, *chikungunya virus* infection, respectively.

## RESULTS

RpNGS offers intuitive web interfaces with summary datasets, analyzing dataset and test report tabs. The primary process of the program is illustrated in Fig. S1.

### Key functionalities

The summary page (Fig. 1) permits users to visually assess the achieved mNGS test. The interactive visualization choice is a map, pie chart, and bar plot to illustrate the sales volume among locations, proportion of each sample type and samples size distribution between months in one year, respectively. RpNGS employ primary data table to display detail information of processed mNGS test for feasible searching and double checking the clinical reports.

In the RpNGS second tab (Fig. S2), the trained experimenter should update the information of each batch including flow cell ID, sample ID, nucleic acids concentration after extraction and library preparation processes, adaptor ID, and file name of sequencing data. Then start the process step by click the process button. There are six steps inside the process pipeline include detecting and copying the sequencing data from sequencer to server, quality control, host reads removal, and classification, abundance estimate, mapped reads extraction. To assist with distinguishing reads from microbes existing in the reagent and lab's environment, RpNGS produced z-scores of taxons in each sample by comparison relative abundance estimations of species to water-only or other control sample collections. The progress log will show the status of five critical operations during data analyze.

Multiple detected pathogens are presented in the candidate pathogen list obtained by analysis process (*Liu et al., 2021*). As a clinical level pathogen detection test, it is

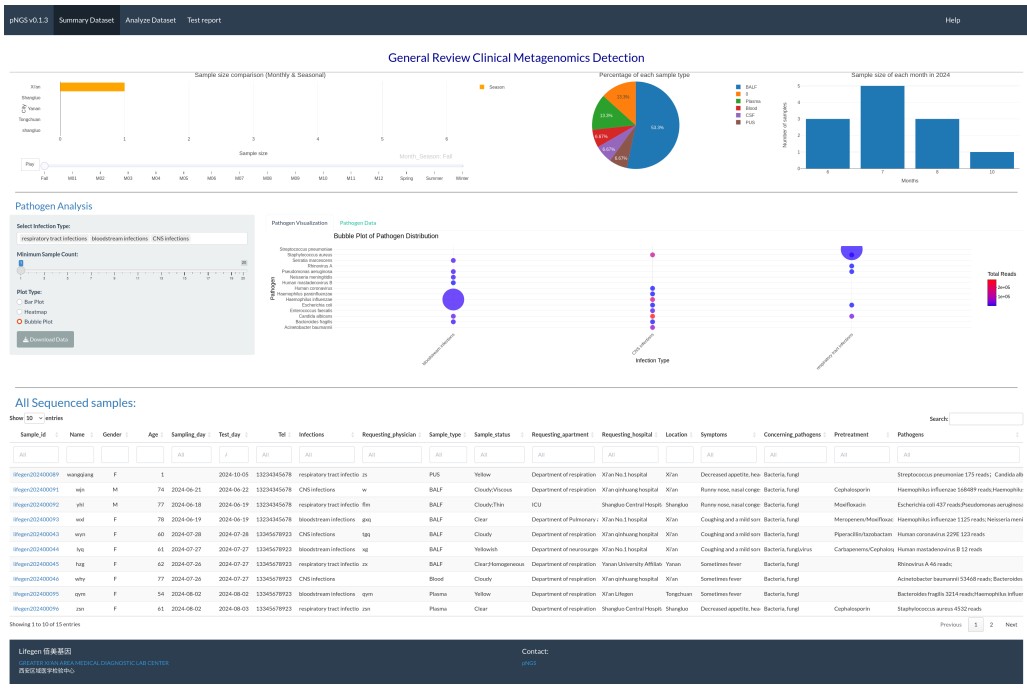

**Figure 1** **Interactive image illustrating sequencing samples, including test volume across various locations, sample types, and test volume per month.** The main table gives full information on these samples, including the report status of each.

important for subset pathogens from the candidate pathogen list depending on their clinical significance. However, identifying whether the detected microbe is the causing pathogen requires a full examination integrating the patient's symptoms, epidemiological context, and other relevant aspects, that should be done by certified clinician, consular or medical doctor. Results are displaced in the third tab (as in Fig. S3) based on user selection of a number of criteria, including z-scores, mapped reads, average coverage for a particular pathogen, gender, age, sample types, clinically relevant pathogen types, and anti-infection treatment, and interface allow user to select the pathogen for a particular patient from the list of microbes. The user should input the flow cell ID for getting test samples in a status table with three columns, flow cell ID, sample ID and status columns. The clinical information and raw microorganisms list for the sample are activated based on the selection in the report status table. To assist with pathogen detection, the background color of microbes indicated in all putative pathogen candidates extracted from 132 clinical metagenomics reports, generated by Shaanxi Lifegen Co., Ltd, are filled with green. The alignment view and related data of specific species will emerge by choosing one of these colored pathogens in raw microorganism list table. Finally, the pathogen table will be formed by repeated selections in the raw microbe's table. An important feature is the ability to export experiment data, patient basic information, confirmed pathogens and related aspects of corresponding pathogen in a Word document as clinical pathogen test report for each sample.

## Pathogens identification pipeline validation

The presence of diverse amounts and types of bacteria, fungi, DNA viruses, RNA viruses, and human cells in the S1–S12 samples, majority of laboratories performed DNA and RNA extraction and sequencing individually for all samples, including NC and S13. The DNA sequencing data were largely utilized used to calculate precision, recall, and F1-score based on the detection of bacteria, fungi, and DNA viruses in each sample, while the RNA sequencing data were used to evaluate the corresponding metrics for RNA viruses. The detection of low-abundance taxa by IDseq may contribute to a drop in precision compared to the precision acquired by RpNGS (Fig. 2).

We also note that low precision (<0.1) among all instruments that caused by significant number of false positive microbes based on precision formula. As shown in Fig. S4, the recall of DNA sequencing data for samples S11, S12, S3, S4, S6, and S9 analyzed by RpNGS are lower than those acquired by IDseq_NR and IDseq_NT. This mismatch is likely due to Bracken's default filtering threshold (Lu et al., 2017), which requires the read count to exceed 10. Compared to IDseq, RpNGS did not detect certain pathogens, as shown in Table S3. For example, in sample S11, Kraken2uniq identified only six reads for *Haemophilus influenzae*, which is comparable with the number of reads allocated to *Haemophilus influenzae* by IDseq_NT. These results indicate that the threshold of taxonomic reads also critical for DNA-based pathogen detection, special for bacteria. For the RNA sequencing data of samples, RpNGS exhibited an equivalent recall rate for RNA virus detection compared to IDseq.

To further remove false positive caused by factors such as reagents and environmental contamination, we employed Z-score normalization to correct the microbial abundance in the samples. With the removal of contaminant-derived false-positive microorganisms, the precision of RpNGS and IDseq in detecting bacteria, fungus, DNA viruses, and RNA viruses improved while maintaining recall rates (Figs. S5, S6). Additionally, this method greatly boosted the F-score of IDseq comparing it with non-correction (Figs. S7, S8).

## Use case: RpNGS for pathogen detection in cases of pediatric meningitis

The RpNGS pipeline was developed for microbial species identification within a human genetic background. However, finding the true causative pathogens of a disease remains problematic due to several influencing factors, including the choice of metagenomic shotgun sequencing analytic methodologies, categorization databases, microbial genetic similarity, and environmental contamination.

To boost pathogen detection, we integrate many components into a single panel, including read categorization tables, clinical metadata, pathogen lists, and associated genome coverage data. To illustrate RpNGS's capability in clinical pathogen identification, we re-evaluated three cerebrospinal fluid (CSF) samples from a study investigating pediatric meningitis in Bangladesh, which were also analyzed using IDseq.

Figure 3 illustrates the production of a clinical report following data analysis section. Clinical information for sample CHRF0002 is given in an editable table, allowing users to alter patient and sample details (Fig. 3A). Users can pick pathogens from the raw microbial

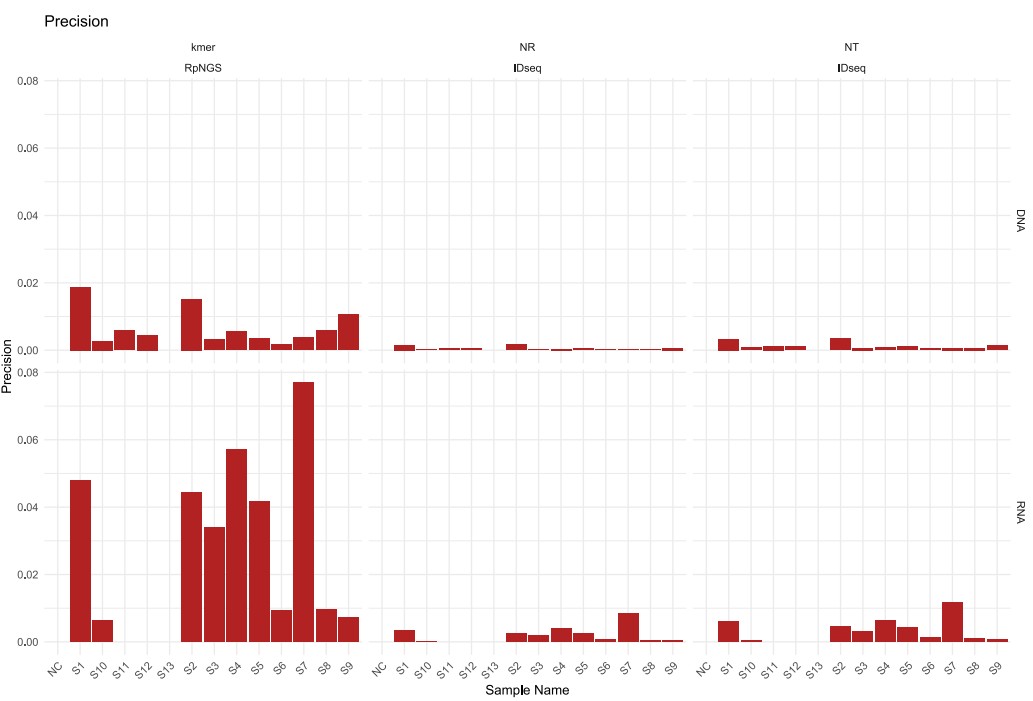

**Figure 2** The precision of IDseq_nt, IDseq_nr and RpNGS for DNA and RNA sequencing datasets among 14 samples.

list depending on clinical significance (Fig. 3B). Microbes with the greatest mapping reads in the raw table, indicated in green according to the specified pathogen list, undergo genome coverage visualization to assess read alignment with reference genomes. For instance, the genome coverage of *Streptococcus pneumoniae* reached 81.52%, increasing the evidence for its pathogenic role (Figs. 3C, 3D).

Additionally, we observed a considerably larger number of reads mapping to *Streptococcus pneumoniae* in CHRF0002 and *Chikungunya virus* in CHRF0094 compared to other detected microorganisms (Fig. 4). Based on these data, we identified *Streptococcus pneumoniae* and *Chikungunya virus* as the primary pathogens for these two samples, which fits with prior studies (Fig. S9, Table S4) (*Saha et al., 2019*; *Kalantar et al., 2020*). These results highlight the amount of categorized reads as critical evidence for infection detection.

## DISCUSSION

RpNGS, developed as a diagnostic tool, was evaluated since the RpNGS clinical metagenomic data analysis pipeline achieved the best F-scores compared with IDseq. It was also the only tool, integrate several criteria into one panel for useful to decide about clinical pathogens and monitor the trend of infection pathogens.

Metagenomic sequencing is a complex process requiring both wet-lab and computer analyses. Various parameters, including sequencing platforms, nucleic acid extraction kits, library preparation procedures, and sequencing strategies, can effect microbial

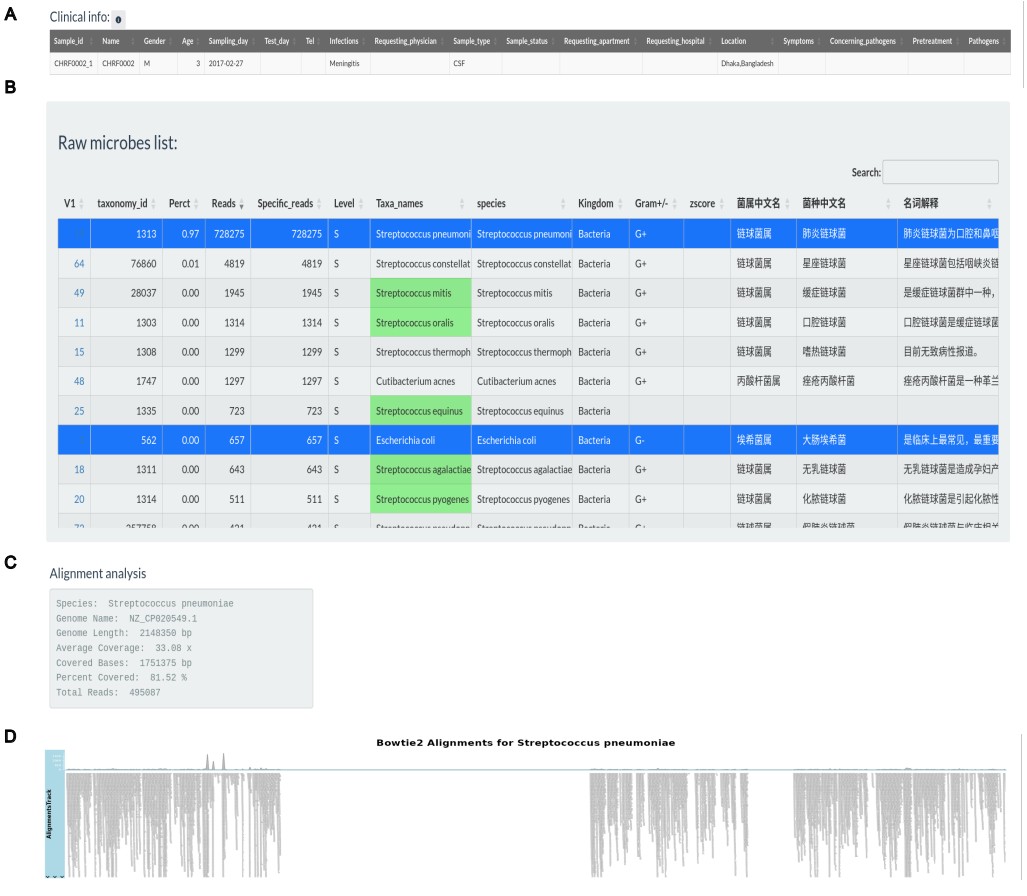

**Figure 3** **Pathogens found in the sample CHRF0002 acquired from patient with pediatric meningitis.**
(A) Clinical info update area. (B) Detected pathogens *via* analytical workflow. (C) The detail info about coverage of *Streptococcus pneumoniae* reference genome. (D) The coverage illustration for the coverage of *Streptococcus pneumoniae* genome.

identification (*Szóstak et al., 2022*). Unlike research-oriented metagenomic data analysis, clinical metagenomic testing must balance cost, turnaround time, and accuracy. While research sequencing techniques typically use paired-end 150 bp reads, clinical sequencing often depends on single-end 50–100 bp reads, with each sample generating approximately 10–20 million reads (*Li et al., 2024*). Moreover, clinical metagenomic testing necessitates creating an infection report identifying possible pathogens within 20 h.

The computational analysis pipeline consists of several essential processes, including quality control, host sequence elimination, taxonomy classification, and pathogen filtering. Each phase offers virous software solutions with comparable functionality, making it vital to determine the most accurate tools for clinical application (*Liu et al., 2021*). To do this, we evaluated the performance of our pipeline using external quality assessment (EQA) data. Additionally, to demonstrate its clinical value, we validated the pipeline using previously sequenced clinical metagenomic datasets.

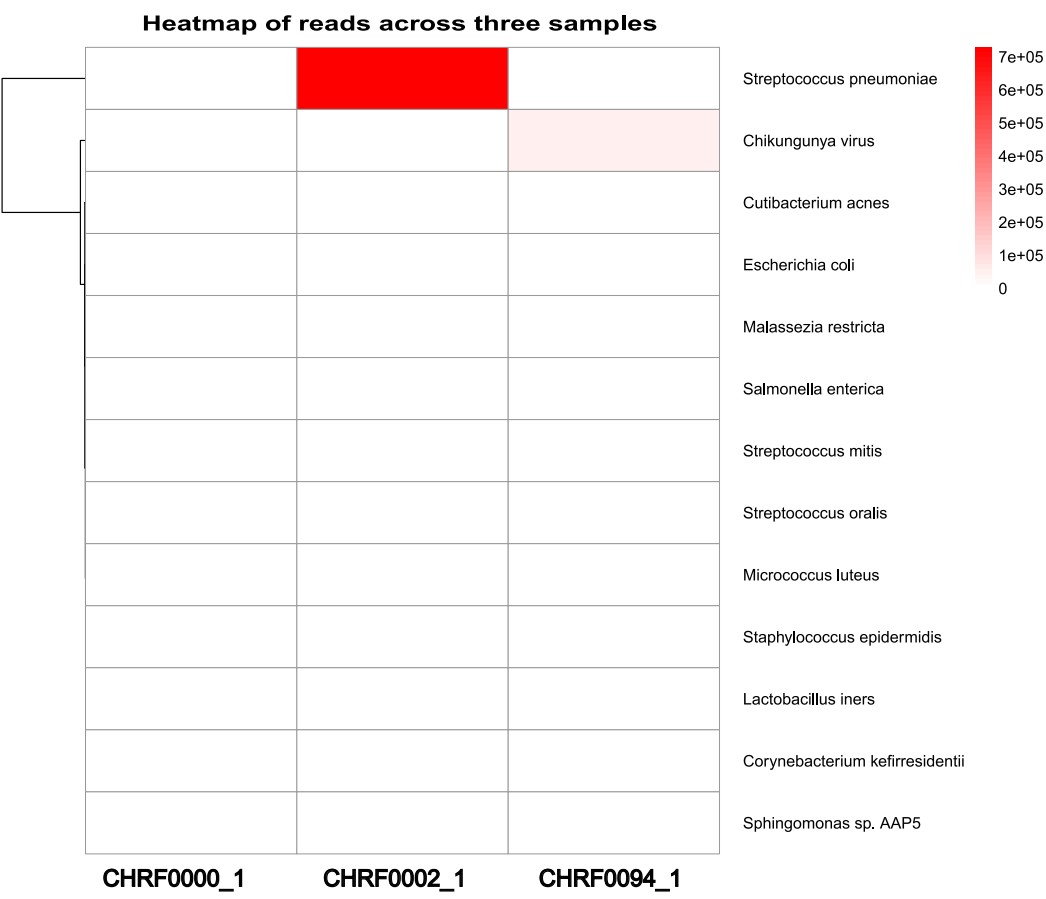

**Figure 4** **Heatmap review of microbials abundance among samples.**

Different taxonomic approaches to classification, such as MetaPhlAn (*Beghini et al., 2021*), Centrifuge (*Govender & Eyre, 2022*) and Kraken2 utilize their own standard databases. However, these standard databases may not necessarily include a comprehensive set of archaea, bacteria, fungus, and viruses by default. For instance, the Centrifuge database (h+p+v+c, released 03/29/2020) lacks fungal species, whereas the Kraken2 standard database (PlusPFP, released 12/28/2024) does not include *Pneumocystis jirovecii*, a clinically significant opportunistic pathogen (*Zhang et al., 2021*). This omission can result in false-negative diagnoses for *Pneumocystis jirovecii* infections. Differences in taxonomic names, particularly for RNA virus, are used in these standard databases (Table S1). Moreover, the microbial composition and abundance in different biological specimens vary significantly, further underlining the importance of developing a customized clinical microbial classification database adapted to specific clinical applications.

False positives are a critical issue in metagenomic data analysis, influenced by both experimental and computational factors. Various methods have been employed to mitigate false positives, including calculating the Z-score for each taxon in sample (*Breitwieser & Salzberg, 2020*), evaluating pathogen genome coverage, reassigning reads using Bracken (*Garrido-Sanz, Senar & Piñol, 2022*), and setting RPM (reads per million)

thresholds based on pathogen type (*Liang et al., 2023*). The use of Z-score can help eliminate false positives introduced by contamination, but its effectiveness depends on the number of control samples available. In RpNGS, *Kraken2uniq* combined with *Bracken* is implemented for microbial classification. Our analysis revealed that while Bracken effectively removes false positives, it can also eliminate low-abundance viruses and fungi. Using an abundance threshold alone only filters out low-abundance false positives, but integrating genome coverage data can further reduce false positives caused by single-region alignments. Additionally, read misassignment between species with high genomic similarity remains a major contributor to false positives in metagenomic detection. For instance, in sample S11, *Haemophilus influenzae* was detected with only six reads, whereas *Haemophilus parainfluenzae* had 21 reads assigned by Kraken2uniq and 23 reads by Bracken, indicating read misclassification. To address this, we propose integrating results from multiple classification algorithms to reduce false positives arising from genomic similarity. Furthermore, removing phage genome sequences from custom-built pathogen detection databases could help minimize false-positive identifications.

## CONCLUSION

RpNGS is a novel open source application that can save clinical fundamental information and experimental data, extract FASTQ data from sequencers and process it, and then manually examine the results to provide a word-type clinical report. Its features enable clinical microbiologists and researcher without bioinformatics or programming expertise to quickly analyze their mNGS data and obtain a deeper comprehension of pathogen detection.

### Funding

This work was supported by the Natural Science Basic Research Program of Shaanxi Province (2021JQ-543), The China Postdoctoral Science Foundation (2020M673605XB), Scientific Research Foundation of Shaanxi University of Science and Technology (2019BT-35), the Shaanxi Qin Chuang Yuan "Scientists + Engineers" Team (2022KXJ-011). The funders had no role in study design, data collection and analysis, decision to publish, or preparation of the manuscript.

### Grant Disclosures

The following grant information was disclosed by the authors:
Natural Science Basic Research Program of Shaanxi Province:  2021JQ-543.
The China Postdoctoral Science Foundation: 2020M673605XB.
Scientific Research Foundation of Shaanxi University of Science and Technology: 2019BT-35.
The Shaanxi Qin Chuang Yuan "Scientists + Engineers" Team: 2022KXJ-011.

### Competing Interests

Ting Hao and Jun Ma are employed by Shaanxi Lifegen Co.

## Author Contributions

- Jing Zhou conceived and designed the experiments, performed the experiments, authored or reviewed drafts of the article, and approved the final draft.
- Yao Tian performed the experiments, analyzed the data, authored or reviewed drafts of the article, and approved the final draft.
- Min Yang performed the experiments, analyzed the data, authored or reviewed drafts of the article, and approved the final draft.
- Ting Hao analyzed the data, prepared figures and/or tables, and approved the final draft.
- Jun Ma analyzed the data, prepared figures and/or tables, authored or reviewed drafts of the article, and approved the final draft.
- Shengyu Wang performed the experiments, analyzed the data, prepared figures and/or tables, authored or reviewed drafts of the article, and approved the final draft.

## Data Availability

Raw data is available at Zenodo:

Ma, J. (2025). RpNGS: An automated platform for pathogen identification and monitoring in clinical metagenomics data [Data set]. Zenodo. https://doi.org/10.5281/zenodo.15469371.

## Supplemental Information

Supplemental information for this article can be found online at http://dx.doi.org/10.7717/peerj.19849#supplemental-information.

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
