# Peer review of "RpNGS: an automated platform for pathogen identification and monitoring in clinical metagenomics data"

_PeerJ, doi:10.7717/peerj.19849_

## Round 0.1 · original submission · Major Revisions

Based on the reviewers’ evaluations, I recommend major revision. Both reviewers acknowledge the practical relevance and potential impact of the RpNGS platform in the field of clinical metagenomics. However, they also raise several important concerns related to language clarity, figure quality, benchmarking completeness, and methodological transparency.

Reviewer #1 emphasized the need for significant improvements in the manuscript’s language, organization of the methods section, and reproducibility of the workflow. The reviewer also requested improved figure resolution and inclusion of tool installation instructions.

Reviewer #2 pointed out grammatical issues, lack of clarity in performance metrics, and recommended including standard benchmarking datasets such as CAMI2 to enhance evaluation robustness.

Both reviewers highlighted the need for better explanation of key parameters and figure annotations. Addressing these issues will substantially improve the clarity and scientific rigor of the manuscript, I believe.

**Language Note:** The review process has identified that the English language must be improved. PeerJ can provide language editing services - please contact us at [email protected] for pricing (be sure to provide your manuscript number and title). Alternatively, you should make your own arrangements to improve the language quality and provide details in your response letter. – PeerJ Staff

Reviewer 1 ·

Basic reporting

Strengths:
• The manuscript addresses a relevant and growing need in clinical microbiology: user-friendly tools for mNGS data interpretation.
• Figures and tables are helpful and relevant.
• Literature cited is largely appropriate and up-to-date.
Areas for Improvement:
• The manuscript contains frequent grammatical and stylistic errors, affecting readability and professionalism. Phrases like “wet data,” “word type clinical report,” and “microbes list” should be revised for clarity.
• The manuscript structure follows acceptable standards, but several sections, such as the software description and analysis workflow, should be reorganized to improve logical flow.
• Some figure captions lack sufficient detail, and several figures (especially Figure 1) are unreadable in their current resolution. High-resolution images should be provided, and complex figures should be split or included as supplementary material.
• The public use of the tool is mentioned, but no clear repository of tool, installation or usage instructions are provided. The authors should include direct links and describe how users can install and run the tool.
• Detailed comments are in the PDF version of the revied manuscript.

Experimental design

Strengths:
• Integration of existing tools (Fastp, Bowtie2, Kraken2, Bracken) into a cohesive, GUI-based workflow is a strong practical contribution.
• Evaluation includes external benchmark datasets and real-world clinical data.
Areas for Improvement:
• Workflow descriptions are somewhat scattered and need better organization. A clear, reproducible step-by-step description of the full pipeline is essential.
• The rationale for key choices (e.g., z-score threshold >1, default Bracken thresholds) should be briefly justified or referenced.
• The manual confirmation process for pathogen detection could be explained why manual curation is required and what is the role of manual curation and add a text about report and certification if author is generating clinical report.

Validity of the findings

Strengths:
• External benchmarking using EQA samples and public mNGS datasets strengthens the manuscript.
• The performance comparison with IDseq is relevant and well-motivated.
Areas for Improvement:
• The results suggest good precision for DNA pathogens but lower recall for RNA viruses. Additional explanation of this tradeoff (e.g., Bracken’s 10-read threshold) is welcome, but further context would help—what is the clinical implication of missing low-abundance viruses?
• The evaluation would benefit from metrics on usability (e.g., runtime, resource usage) and user experience (especially for non-bioinformaticians).

Additional comments

This manuscript presents an important contribution to the field of clinical metagenomics by making advanced sequencing analysis accessible to users without programming expertise. The integration of widely accepted tools into a Shiny-based platform, with automated reporting, addresses a significant barrier in clinical adoption of mNGS. I recommend the author to cite the workflow or tools if the workflow is publicly available as cited in the manuscript.
To improve the manuscript:
• A major revision of language is needed.
• Clarify and structure the methods more systematically, particularly the sequencing analysis pipeline and benchmarking process.
• Improve figure resolution and enhance legend clarity.

Annotated reviews are not available for download in order to protect the identity of reviewers who chose to remain anonymous.

·

Basic reporting

Multiple grammar issues:
1. With the broadly and successfully application in clinical pathogen detection.. broadly and successfully -> broad and successful
2. The IDseq was selected because its well-known cloud-based, open-source bioinformatics platform developed for the clinical metagenomic analysis. Its -> it's; the clinical -> clinical
3. RpNGS offers intuitive web interfaces with summary dataset, analyze dataset and test report tabs. dataset-> datasets; analyze dataset -> analyzing dataset
4. calculated z-scores of taxons in each sample by compared relative abundance estimations of specie to water-only or other control sample collections. specie->species; compared-> comparing

There are several more grammar issues, I suggest the authors check the draft thoroughly again.

Experimental design

The authors present a new R-based application—RpNGS—for the analysis of clinical metagenomic sequencing data.
The authors integrated several open source tools such as fastq, bowtie2 and kraken. The authors introduced the interface thoroughly with figures.

I find the program useful, and the experimental design is somewhat solid. However, in the meantime I think there are several points that need improvement:
1. While the authors benchmark RpNGS against synthetic microbial mixtures and clinical datasets, the evaluation omits widely used benchmarking resources such as the Critical Assessment of Metagenome Interpretation (CAMI2) dataset. Including results from CAMI2 would provide a more standardized and comprehensive assessment of the pipeline's taxonomic resolution, detection limits, and comparative performance, particularly since CAMI2 is specifically designed for evaluating metagenomic classification tools under complex and realistic scenarios. I recommend the authors consider incorporating CAMI2 into their benchmarking suite to strengthen the evaluation and facilitate comparisons with other established methods.

2. Clarify Low Precision Values in Figure 2A
The precision scores reported in Figure 2A are unexpectedly low (often <0.1) across all tools, including RpNGS. The manuscript does not adequately explain why such low precision is observed. The authors should clarify whether this is due to high false positive rates, limitations in reference databases, permissive thresholds, or characteristics of the benchmark dataset. Without this context, it's difficult to interpret the practical diagnostic value of the tool.

Validity of the findings

The conclusions are generally well supported by the results. However, the authors should more clearly define the novelty and clinical impact of RpNGS relative to existing tools. Additionally, the lack of statistical analysis and absence of error estimates in benchmarking (e.g., Figure 2) limits the confidence in the comparative claims. Finally, the manuscript would benefit from a clearer rationale for how RpNGS could be adopted or extended by other clinical labs.

Additional comments

Minor comments:
1. Image on github is broken.
2. Word-format reports is bad to interpret. I suggest using JSON/TSV format.
3. Figures are generally having too small fonts.
4. Figure 2 and 3 need to rework. The authors need to The x-axis seems to represent individual samples, but they’re unlabeled. Without sample identifiers (e.g., S1, S2…), it’s hard to link figure trends with text commentary (like “RpNGS shows lower recall in S11”). Grouping DNA vs. RNA samples visually (e.g., color blocks or labels) would also help. For figure 3 the the color bar need to be marked more clearly. Especially when there are tree branches with white blocks that I cannot tell the difference.

---

## Round 0.2 · Minor Revisions

Based on the reviewer’s feedback, I recommend a minor revision. The reviewer acknowledged the overall soundness of the work and did not raise any concerns regarding the methodology, analysis, or conclusions. The suggested revisions focus primarily on minor typographical errors and rewording of certain sentences to improve clarity and readability for the audience. Addressing these points will enhance the presentation and accessibility of the manuscript.

Reviewer 1 ·

Basic reporting

Strengths:
• The manuscript addresses a relevant and growing need in clinical microbiology: user-friendly tools for mNGS data interpretation.
• Figures and tables are helpful and relevant.
• Literature cited is largely appropriate and up-to-date.
Areas for Improvement:
• Detailed comments are in the PDF version of the reviewed manuscript.

Experimental design

Experimental design
Strengths:
• Integration of existing tools (Fastp, Bowtie2, Kraken2, Bracken) into a cohesive, GUI-based workflow is a strong practical contribution.
• Evaluation includes external benchmark datasets and real-world clinical data.

Validity of the findings

Strengths:
• External benchmarking using EQA samples and public mNGS datasets strengthens the manuscript.
• The performance comparison with IDseq is relevant and well-motivated.

Additional comments

Additional comments
This manuscript presents an important contribution to the field of clinical metagenomics by making advanced sequencing analysis accessible to users without programming expertise. The integration of widely accepted tools into a Shiny-based platform, with automated reporting, addresses a significant barrier in clinical adoption of mNGS. I recommend the author(s) to review the feedback and make correction because some are typo and some are rewording the sentence to make it clear to the reader.
To improve the manuscript:
• A minor revision of language is needed but does not require secondary review before publication.

Good luck to the authors for publication!

Annotated reviews are not available for download in order to protect the identity of reviewers who chose to remain anonymous.

·

Basic reporting

I have no further questions.

Experimental design

I have no further questions.

Validity of the findings

I have no further questions.

---

## Round 0.3 · accepted · Accept

The authors have thoroughly and satisfactorily addressed all reviewer comments. The manuscript is now suitable for acceptance in its current form.